# Gait Analysis with Wearables Is a Potential Progression Marker in Parkinson’s Disease

**DOI:** 10.3390/brainsci12091213

**Published:** 2022-09-08

**Authors:** Sha Zhu, Zhuang Wu, Yaxi Wang, Yinyin Jiang, Ruxin Gu, Min Zhong, Xu Jiang, Bo Shen, Jun Zhu, Jun Yan, Yang Pan, Li Zhang

**Affiliations:** 1Department of Geriatric Neurology, The Affiliated Brain Hospital of Nanjing Medical University, Nanjing 210029, China; neuro_zhusha@stu.njmu.edu.cn (S.Z.); wangyaxi1102@stu.njmu.edu.cn (Y.W.); neuo_jiangyinyin@stu.njmu.edu.cn (Y.J.); grx@njmu.edu.cn (R.G.); neuro_zhongmin@163.com (M.Z.); jx13773958367@sina.com (X.J.); 15951758927@163.com (B.S.); njmuzhangli@sina.com (J.Z.); kdzb666@163.com (J.Y.); neuro_panyang@163.com (Y.P.); 2Neurotoxin Research Center of Key Laboratory of Spine and Spinal Cord Injury Repair and Regeneration of Ministry of Education, Neurological Department of Tongji Hospital, School of Medicine, Tongji University, Shanghai 200092, China; 2111723@tongji.edu.cn

**Keywords:** Parkinson’s disease, walking, gait analysis, wearable sensors

## Abstract

Gait disturbance is a prototypical feature of Parkinson’s disease (PD), and the quantification of gait using wearable sensors is promising. This study aimed to identify gait impairment in the early and progressive stages of PD according to the Hoehn and Yahr (H–Y) scale. A total of 138 PD patients and 56 healthy controls (HCs) were included in our research. We collected gait parameters using the JiBuEn gait-analysis system. For spatiotemporal gait parameters and kinematic gait parameters, we observed significant differences in stride length (SL), gait velocity, the variability of SL, heel strike angle, and the range of motion (ROM) of the ankle, knee, and hip joints between HCs and PD patients in H–Y Ⅰ-Ⅱ. The changes worsened with the progression of PD. The differences in the asymmetry index of the SL and ROM of the hip were found between HCs and patients in H–Y Ⅳ. Additionally, these gait parameters were significantly associated with Unified Parkinson’s Disease Rating Scale and Parkinson’s Disease Questionnaire-39. This study demonstrated that gait impairment occurs in the early stage of PD and deteriorates with the progression of the disease. The gait parameters mentioned above may help to detect PD earlier and assess the progression of PD.

## 1. Introduction

Parkinson’s disease (PD) is a progressive neurodegenerative disease characterized primarily by movement disorders [1]. Gait dysfunction is one of the primary motor symptoms in PD [2], which impacts quality of life and increases the risk of falling [3,4]. At present, the early diagnosis of PD remains challenging for neurologists [5]. Reliable biomarkers that can detect PD at an earlier stage are needed to intervene and monitor potential disease-modifying therapies. In addition, the clinical symptoms may develop heterogeneously during the progression of PD. The existence of objective markers sensitive to long-term and short-term changes has implications for the evaluation and adjustment of therapeutic interventions. Therefore, it is necessary to explore objective, sensitive, and reliable methods to obtain markers of PD. 

Gait parameters may have the potential to be markers of PD. Individuals with PD tend to show a decrease in step length and walking speed, as well as an increase in step time [6]. Previous gait assessments of PD patients were limited to clinician observations and scale evaluations [7]. Although the Unified Parkinson’s Disease Rating Scale part 3 (UPDRS III) score is widely used to assess motor function in PD, it has intra- and inter-rater reliability issues and only partially reflects motor function in daily life [8,9]. Given the complexity of gaits, especially when their changes are subtle, they can be difficult to capture with the naked eye. With the development of technology, gait parameters derived from instrumented motion-analysis systems may allow walking patterns to be quantified. However, the utility of this sensing technique is often limited to laboratory settings. Stereophotogrammetry based on an optoelectronic sensor is considered to be the gold standard, but is costly and requires a controlled and specialized movement environment [10]. Wearable devices are small, lightweight sensors such as inertial measurement units that are attached to one or more body parts. Wearable sensors are suitable for gait analysis in daily life, providing a means to objectively provide personalized gait characteristics [11,12]. However, the current research on the use of wearable sensor technology in the clinical assessment of PD patients is still lacking, especially on kinematic gait parameters.

The Hoehn and Yahr (H–Y) scale is the most commonly used tool for determining the degree of PD progression through simple staging [13,14]. According to the H–Y scale, PD patients had the worst walking quality in the most advanced stages of the disease, but limited information is available on the evolution of gait parameters. Although some studies have analyzed the gait patterns of PD, few studies have reported on kinematic parameters and symmetry parameters according to different H–Y stages [15]. Moreover, a consensus is needed on which gait values are relevant. This study analyzed the walking pattern (spatiotemporal, kinetic, and symmetry parameters) in PD patients based on the evolutionary stage (Ⅰ–Ⅱ, Ⅲ and Ⅳ) as defined by the H–Y scale and investigated the correlation between gait parameters and clinical scales. We aimed to explore whether these gait parameters can be used as markers of PD, and to determine which parameters can help in the early detection of PD and which can be correlated with PD progression.

## 2. Method

### 2.1. Patients

From February 2019 to July 2021, 138 patients with PD were recruited from the Department of Geriatrics, Nanjing Brain Hospital Affiliated with Nanjing Medical University. All patients were diagnosed with PD according to the Movement Disorder Society (MDS) criteria [16] and they could understand and respond to the doctor’s instructions. We excluded patients with other diseases that can affect gait performance, such as cerebrovascular disease, fractures, and spinal spectrum diseases, etc. We also enrolled 56 healthy controls (HCs) from the patients’ escorts and hospital staff. The Medical Ethics Committee of the Nanjing Brain Hospital Affiliated with Nanjing Medical University reviewed and approved this research. All participants signed written informed consent before the study.

### 2.2. Clinical Evaluation

In our research, we collected the following demographic characteristics of all participants: age, gender, weight, height, degree of education, shoe size, and duration of the disease. The H–Y scale was applied to assess the disease disability of PD patients. UPDRS part 1 (UPDRS Ⅰ) was used to assess mental, behavioral and mood, UPDRS part 2 (UPDRS Ⅱ) was used to evaluate activities of daily living, and UPDRS Ⅲ was applied to assess the severity of motor symptoms. The quality of life was assessed by Parkinson’s Disease Questionnaire-39 (PDQ39). A total of 138 PD patients were assessed by H–Y scales in the “off” medication state and were divided into three groups: H–Y Ⅰ–Ⅱ (64 individuals), H–Y Ⅲ (53 individuals) and H–Y Ⅳ (21 individuals). 

### 2.3. Instruments

A gait-analysis system called JiBuEn was used in our study to collect gait data. Previous research has proven the accuracy of this device [17]. The JiBuEn gait-analysis system consists of a Bluetooth module fixed under smart shoes and modules with inertial micro-electro-mechanical system sensors. Moreover, it included 4 modules tied to the subject’s thigh and calf, and 1 module tied to the subject’s waist (Figure 1). Through wireless Bluetooth transmission technology, the gait data were transmitted into the computer system in real time and the final gait data were obtained. 

### 2.4. Gait Data Collection

All PD patients stopped using anti-PD drugs for 24 h (the controlled-release anti-PD drug was 72 h). Gait data were collected the next morning in a fully awake state. All subjects were instructed to complete the instrumented stand-and-walk test [18]. Participants firstly stood quietly for 30 s, and when they heard the doctor’s instruction, they began to walk for 7 m in a free manner, and then turned back to the initial place. Doctors explained the procedure of the test in detail to the participants before the test. In addition, all participants walked twice in advance to familiarize themselves with the test. In this process, gait data were collected.

### 2.5. Statistical Analysis

Data were analyzed by statistical package SPSS v. 25.0. The chi-squared test was used for qualitative data. The Kolmogorov–Smirnov test was initially used to check whether the measurement data followed a normal distribution. For normally distributed data, the one-way ANOVA was used. For non-normally distributed data, the Kruskal–Wallis H-test was used. The Bonferroni correction was followed for multiple comparisons. A Spearman correlation analysis was performed to investigate the association between gait parameters and H–Y stage, UPDRS scores, and PDQ39. Data were displayed as the mean ± standard deviation (SD), and *p* < 0.05 was considered to be significant. We used Formula (1a) to calculate the variability of the gait parameters, and then used Formula (1b) to combine them [19,20]. We used the asymmetry index (AI) to evaluate the asymmetry of the gait parameters through Formula (2) [21,22]. We also used Formula (3) to calculate the walk ratio [23,24].
CV_separate_ = SD ÷ mean value(1a)

(1b)
$${\mathrm{CV}}_{\mathrm{combined}}=\;\sqrt{\frac{{\mathrm{CV}}_{\;L}+{\;\mathrm{CV}}_{\;R}}{2}}\;\times \;100$$


The subscripts L and R represent the left and right sides of the individual, respectively. CV stands for coefficient of variation

(2)
$$\mathrm{AI}=\frac{\;\max\;\left(X_{L\;},{\;X}_{R}\right)-\min\;\left(X_{L\;},{\;X}_{R}\right)}{\max\;\left(X_{L},{\;X}_{R}\right)}\;\times \;100$$

where X refers to stride length, stride time, stance phase time, swing phase time, heel strike angle, toe-off angle, and the range of motion (ROM) of the ankle, knee, and hip joints
Walk ratio = step length ÷ cadence(3)

## 3. Results

### 3.1. Clinical Baseline Data of Participants

In our research, we enrolled 138 PD patients and 56 HCs. A total of 64 of the patients were in the H–Y Ⅰ–Ⅱ stage, 53 in the H–Y Ⅲ and 21 in the H–Y Ⅳ stage. Their clinical characteristics are shown in Table 1. No statistical difference was found in clinical baseline data between four groups.

### 3.2. Spatiotemporal Gait Parameters

The spatiotemporal gait parameters collected in our research included stride length (SL), cadence (CA), stride time (ST), gait velocity (GV), stance phase time (StPT) and swing phase time (SwPT). The variabilities of SL (CV–SL), ST (CV–ST), StPT (CV–StPT), and SwPT (CV–SwPT) were also collected. No difference was observed between the four groups in terms of CA, ST or StPT(%). However, we observed a remarkable difference in terms of SL, GV, SwPT(%), CV–SL, CV–ST, CV–StPT, CV–SwPT and the walk ratio. Furthermore, in the post-hoc analyses, we found statistically significant differences in terms of SL, GV, and CV–SL between HC, H–Y I–II, and H–Y III. CV–ST was significantly increased in the advanced stage of the disease. We also observed a statistical difference in CV–SwPT and the walk ratio between H–Y I–II and H–Y III. The changes in SwPT(%) and CV–StPT between different stages were modest. All the data are listed in Figure 2.

### 3.3. Kinematic Gait Parameters

We evaluated the kinematic gait parameters by collecting the data of range of motion (ROM) of the ankle (ROM–AJ), knee (ROM–KJ), and hip joints (ROM–HJ). We obtained the ROM value by calculating the difference between the maximum and minimum angles of the three joints mentioned above in the sagittal plane. The heel strike (HS) and toe-off (TO) angles were also included in our research. ROM–HJ and HS were statistically different when comparing the four groups in pairs. We also found significant differences in ROM–AJ and ROM–KJ between HC, H–Y I–II and H–Y III. However, we found no difference in TO between the four groups. All the data are listed in Figure 3.

### 3.4. Symmetry Analysis of Gait Parameters

The spatiotemporal and kinematic gait parameters were studied in the symmetry analysis of the gait parameters. Differences in AI–SL and AI–ROM–HJ were found between HC and patients in H–Y Ⅳ. All the data are shown in Table 2.

### 3.5. Correlation Analysis between Gait Parameters and H–Y Stage, UPDRS Scores and PDQ39

We performed a Spearman correlation analysis between the gait parameters and the H–Y stage, UPDRS Ⅰ, UPDRS Ⅱ, UPDRS Ⅲ, and PDQ39. Considering that AI changes slightly with disease progression according to the H–Y scale, we only included spatiotemporal gait parameters and kinematic gait parameters in the correlation analysis. Table 3 shows that the gait parameters are significantly related to H–Y stage, UPDRS I, UPDRS II, UPDRS III and PDQ39, among which CA and ST are only related to UPDRS I, but not to the other scales.

## 4. Discussion

Gait impairment is a significantly disabling symptom of PD that imposes a serious burden on society and families. In our research, we included 56 HCs and 138 patients with PD. We also divided patients with PD into the following three groups according to the H–Y scale: H–Y Ⅰ–II, H–Y III, and H–Y IV. By using wearable sensors, our research quantified gait parameters in 138 patients with PD at different disease stages. We analyzed the changes in gait performance depending on the stage of PD. This cross-sectional, observational study may provide assistance for clinical and rehabilitation treatment.

### 4.1. Spatiotemporal Gait Parameters

In our research, SL, GV, and CV–SL were significantly worse in early-stage PD patients than in HC, and, importantly, continuously progressed with the development of PD. However, the differences were not pronounced between H–Y III and Ⅳ. SL was regarded as the most prominent gait parameter in patients with PD [25]. Previous studies have shown that a shorter SL is mainly due to the weakened ability to propel the body forward [26], and a shorter SL often implies damage to the balance function [27]. Additionally, a shorter SL is thought to be associated with cognitive impairment [28]. From this indicator, balance and cognitive function are impaired in the early stage of PD. Another study also found changes in SL as PD progressed, but they did not find changes in SL in the early stages of PD, which may be related to their relatively small sample sizes [29]. GV also deteriorates significantly as PD progresses, which also reflects the characteristics of bradykinesia in PD patients. A previous study has shown that GV is related to category fluency, processing speed, and memory [30]. A decrease of 0.1 m/s in habitual GV can be rated as clinically meaningful in terms of health problems and/or therapeutic interventions [31,32]. Meanwhile, the average value of GV in our study decreased by more than 0.1 m/s in each group with the progress of PD. This result suggests that specific rehabilitation and drug interventions for the improvement of GV in PD patients are necessary. CV refers to the stability between steps during the walking process. CV has been reported to be associated with the severity of the disease [33]. Previous reports have suggested that PD patients with a positive history of falls have greater gait variability than non-fallers [34]. This result may help facilitate patients and doctors to intervene and adjust treatment before falls occur. 

CA, StPT and ST did not change in the course of PD, and this finding is not surprising given that many previous studies showed that CA does not usually change in PD [35], or increase as a compensatory mechanism for the reduction in SL [36]. SwPT was reported to decrease in PD patients compared with HCs [37], but only a slight difference was found in SwPT between H–Y III and HCs in our research. In general, the above-mentioned parameters had the lowest ability to discriminate PD patients from HCs and assess disease progression compared with the other parameters. In addition, we did not include step width in our study, and it is generally considered that there is no difference between PD patients and HCs [37].

### 4.2. Kinematic Gait Parameters

Kinematic gait parameters also possess potential as a progression marker in the early stage of PD. We observed a decrease in HS throughout the disease. There was no significant difference in TO between PD patients and HCs, which is in line with a recent study [38]. A shuffling gait is one of the typical features commonly observed in PD and increases the risk of falls in PD patients. The decline in HS means that lifting the lower limbs is difficult during the walking process, which reflects the characteristics of the patient’s shuffling gait from a kinematic point of view. Our research showed that shuffling gait deteriorates with the progression of disease. This result suggests that interventions for patients’ risk of falls should run through the entire course of the disease. 

Angular measurements of the joints of the lower limbs are also related to the characterization of the walking pattern, and only a few studies have previously analyzed these parameters. We found that early-stage PD patients experience a reduced ROM of joints. The damage also gradually worsens as the disease progresses; ROM–HJ especially shows a continuous decrease throughout the disease. A smaller ROM–HJ value indicates a more rigid gait pattern. Previous studies have suggested that the decline in ROM–AJ suggests a decrease in foot clearance [39]. DiPaola et al. discovered that ROM–KJ influences the pendular mechanism of walking [40]. A study that was followed up 10 weeks after rehabilitation showed that the ROM of the joints of PD patients can be improved after rehabilitation [41]. Therefore, rehabilitation for PD patients should be conducted as early as possible.

The symmetry analysis of gait parameters in PD is characterized by the persistent asymmetry of motor symptoms [42]. Gait asymmetry in PD is associated with uncoordinated activity in the leg muscles [43]. Although AI shows an increasing trend with the progress of PD, we discovered that AI plays a relatively minor role as a biomarker of PD progression. We only found a difference in AI–SL and AI–ROM–HJ between advanced-stage PD and HCs. This finding is similar to a previous study, which revealed that gait symmetry remained preserved in de novo drug-naive PD patients [25]. It was previously reported that gait asymmetry may be an early sign of PD [44], and this finding contradicts our results. However, the number of people who converted to PD in this study was relatively small. Possible reasons for the relatively symmetrical gait parameter in our early-stage PD patients may be that the supplementary motor cortex and the motor cortical structures retain symmetrical gait function, which compensates for the asymmetric input from the subcortical structure. However, considering that our research was a cross-sectional study, more prospective longitudinal studies are needed.

UPDRS and PDQ39 can track the progression of PD and are related to the patient’s quality of daily life [45,46,47]. In our study, the gait parameters derived from wearables were significantly correlated with these subjective scales, demonstrating the reliability of quantifying disease progression using wearables. These gait parameters can serve as objective outcome markers of drug therapy and other interventions. A previous wearable-device-based study has also shown that motor measures are highly correlated with daily living and quality of life [48]. Therefore, studies based on these wearable sensors may provide results that benefit PD patients. 

Our research also faces some limitations. Firstly, the number of PD patients in the H–Y IV stage in our research was relatively small, and this study was only conducted in a single center, which may have led to a selection bias. Secondly, PD is a heterogeneous disorder. Other non-motor symptoms and subgroups that may affect gait, such as depression, were not considered in this study. Finally, our study only provides evidence from a cross-sectional protocol; thus, this result should be validated in more prospective longitudinal studies with larger groups of patients.

## 5. Conclusions

In conclusion, despite the limitation we mentioned above, our study analyzed the gait pattern related to the evolutionary stages I–II, III, and IV according to the H–Y scale in patients with PD. This study shows that gait impairment in PD patients runs through the entire course of the disease and occurs even in the early stage of the disease. The SL, GV, CV–SL, ROM of the joints and HS may help to detect PD earlier and assess PD progression. Furthermore, these variables should be the target of rehabilitation and exercise therapies. Conducting longitudinal studies and implementing these measures in clinical trials can be the next steps in further assessing the efficacy of these parameters.

## Figures and Tables

**Figure 1 brainsci-12-01213-f001:**
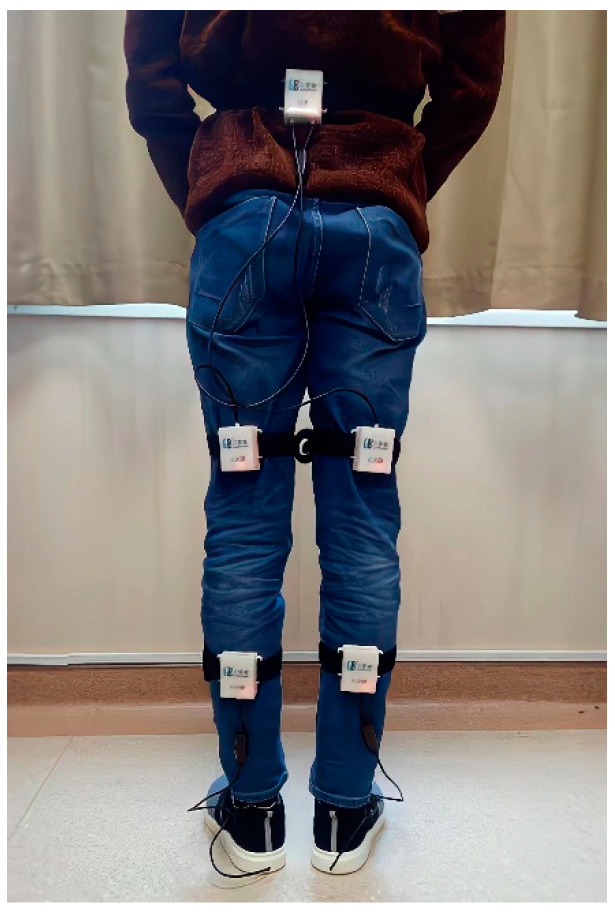
Photograph of JiBuEn gait-analysis system.

**Figure 2 brainsci-12-01213-f002:**
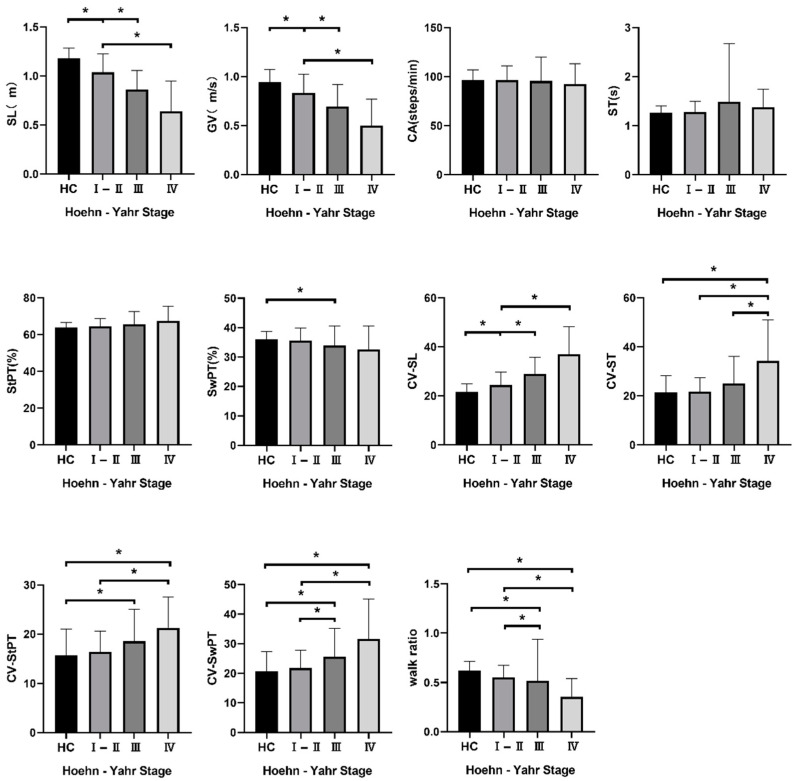
Spatiotemporal gait parameters of participants. Note: Value is listed as mean ± SD; * *p* < 0.05. Abbreviations: PD: Parkinson’s disease; HC: healthy control subjects; H–Y stage: Hoehn–Yahr stage; SL: stride length; ST: stride time; GV: gait velocity; CA: cadence; CV: coefficient of variation; StPT: stance phase time; SwPT: swing phase time.

**Figure 3 brainsci-12-01213-f003:**
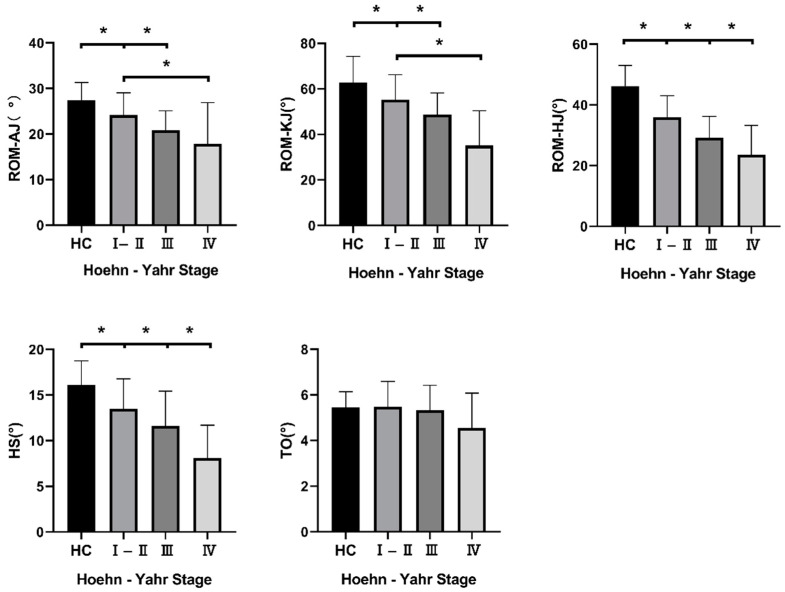
Kinematic gait parameters of participants. Note: Value is listed as mean ± SD; * *p* < 0.05. Abbreviations: PD: Parkinson’s disease; HC: healthy control subjects; H–Y stage: Hoehn–Yahr stage; ROM: range of motion; AJ: ankle joint; KJ: knee joint; HJ: hip joint; HS: heel strike angle; TO: toe-off angle.

**Table 1 brainsci-12-01213-t001:** Clinical baseline data of patients.

	HC	H–Y Ⅰ–Ⅱ	H–Y Ⅲ	H–Y Ⅳ	*p*
N	56	64	53	21	
Age (years)	62.36 ± 6.60	65.61 ± 9.47	66.66 ± 10.44	65.43 ± 6.90	0.066
Height (cm)	163.36 ± 5.98	165.20 ± 6.61	163.70 ± 8.23	163.57 ± 7.89	0.536
Weight (kg)	63.02 ± 8.37	65.99 ± 10.65	65.47 ± 11.86	64.50 ± 9.41	0.386
Male (%)	28 (50%)	41 (64.1%)	26 (49.1%)	8 (38.1%)	0.136
Education (%)					0.360
Illiteracy	2 (3.6%)	6 (9.4%)	2 (3.8%)	2 (9.5%)	
Primary school	9 (16.1%)	10 (15.6%)	9 (17.0%)	3 (14.3%)	
Middle school	41 (73.2%)	35 (54.7%)	35 (66.0%)	15 (71.4%)	
College	4 (7.1%)	13 (20.3%)	7 (13.2%)	1 (4.8%)	
Shoes size	39.18 ± 1.89	39.97 ± 2.09	39.66 ± 2.06	39.71 ± 2.08	0.213
PD duration (years)		3.51 ± 3.73	8.13 ± 7.69	10.14 ± 3.93	
UPDRS III total scores		22.77 ± 9.24	35.72 ± 11.81	55.14 ± 15.93	

Note: Value is shown as mean ± SD. Abbreviations: PD: Parkinson’s disease; HC: healthy control subjects; H–Y stage: Hoehn–Yahr stage; UPDRS III: Unified Parkinson’s Disease Rating Scale part 3.

**Table 2 brainsci-12-01213-t002:** Symmetry analysis of gait parameters.

	HC	H–Y Ⅰ–Ⅱ	H–Y Ⅲ	H–Y Ⅳ	*p*	*Post-Hoc*	
AI–SL	2.29 ± 0.95	2.57 ± 1.41	3.05 ± 1.92	5.39 ± 4.87	**<0.001**	**<0.001 ^1^**	**0.002 ^2^**
AI–ST	7.38 ± 8.62	9.34 ± 9.35	11.64 ± 15.53	13.82 ± 12.31	0.087		
AI–StPT (%)	5.51 ± 6.29	5.37 ± 5.17	7.80 ± 12.82	7.53 ± 9.58	0.794		
AI–SwPT (%)	8.94 ± 8.89	9.21 ± 8.10	11.62 ± 13.97	13.66 ± 11.85	0.351		
AI–HS	14.27 ± 9.66	16.89 ± 12.26	16.96 ± 11.98	21.84 ± 13.29	0.125		
AI–TO	8.84 ± 7.04	10.42 ± 8.76	11.33 ± 12.09	11.88 ± 8.08	0.445		
AI–ROM–AJ	9.59 ± 7.35	10.95 ± 8.48	14.17 ± 11.58	10.87 ± 10.84	0.131		
AI–ROM–KJ	13.49 ± 10.43	16.22 ± 13.49	18.44 ± 14.62	16.54 ± 17.25	0.338		
AI–ROM–HJ	8.84 ± 7.12	12.00 ± 9.81	13.82 ± 15.94	18.68 ± 14.81	**0.045**	**0.030 ^1^**	

Note: Value is listed as mean ± SD, bold font indicates significant results. Abbreviations: PD: Parkinson’s disease; HC: healthy control subjects; H–Y stage: Hoehn–Yahr stage; AI: asymmetry index; ST: stride time; SL: stride length; StPT: stance phase time; SwPT: swing phase time; TO: toe-off angle; HS: heel strike angle; AJ: ankle joint; KJ: knee joint; HJ: hip joint; ROM: range of motion. ^1^ Comparisons of variables between H–Y Ⅳ and HC; ^2^ Comparisons of variables between H–Y Ⅳ and H–Y Ⅰ–Ⅱ.

**Table 3 brainsci-12-01213-t003:** Correlation analysis between gait parameters and H–Y stage, UPDRS scores and PDQ39.

	H–Y Stage	UPDRS I	UPDRS II	UPDRS III	PDQ39
SL (m)	**−0.566 (<0.001)**	−0.155 (0.070)	**−0.453 (<0.001)**	**−0.454 (<0.001)**	**−0.433 (<0.001)**
GV (m/s)	**−0.458 (<0.001)**	−0.040 (0.644)	**−0.334 (<0.001)**	**−0.376 (<0.001)**	**−0.269 (0.001)**
CA (steps/min)	−0.052 (0.545)	**0.212 (0.012)**	0.001 (0.987)	−0.069 (0.421)	0.110 (0.201)
ST (s)	0.055 (0.518)	**−0.210 (0.013)**	0.003 (0.970)	0.075 (0.383)	−0.109 (0.204)
StPT (%)	**0.247 (0.003)**	−0.073 (0.396)	**0.199 (0.019)**	**0.229 (0.007)**	0.092 (0.284)
SwPT (%)	**−0.268 (0.001)**	0.057 (0.504)	**−0.216 (0.011)**	**−0.253 (0.003)**	−0.062 (0.469)
CV−SL	**0.552 (<0.001)**	**0.219 (0.010)**	**0.500 (<0.001)**	**0.463 (<0.001)**	**0.358 (<0.001)**
CV−ST	**0.401 (<0.001)**	**0.194 (0.023)**	**0.369 (<0.001)**	**0.371 (<0.001)**	**0.224 (0.008)**
CV−StPT	**0.380 (<0.001)**	0.153 (0.074)	**0.315 (<0.001)**	**0.311 (<0.001)**	**0.229 (0.007)**
CV−SwPT	**0.455 (<0.001)**	0.107 (0.211)	**0.373 (<0.001)**	**0.346 (<0.001)**	**0.252 (0.003)**
walk radio	**−0.431 (<0.001)**	**−0.263 (0.002)**	**−0.380 (<0.001)**	**−0.343 (<0.001)**	**−0.441 (<0.001)**
ROM−AJ (°)	**−0.382 (<0.001)**	**−0.250 (0.003)**	**−0.287 (0.001)**	**−0.281 (0.001)**	**−0.375 (<0.001)**
ROM−KJ (°)	**−0.429 (<0.001)**	−0.107 (0.210)	**−0.327 (<0.001)**	**−0.340 (<0.001)**	**−0.434 (<0.001)**
ROM−HJ (°)	**−0.517 (<0.001)**	−0.118 (0.168)	**−0.408 (<0.001)**	**−0.422 (<0.001)**	**−0.332 (<0.001)**
HS (°)	**−0.454 (<0.001)**	−0.133(0.121)	**−0.398 (<0.001)**	**−0.379 (<0.001)**	**−0.246 (0.004)**
TO (°)	**−0.171 (0.044)**	−0.089 (0.302)	−0.105 (0.220)	**−0.144 (<0.001)**	**−0.067 (0.434)**

Notes: Data are listed as r (*p*); bold font means significant results. Abbreviations: SL: stride length; GV: gait velocity; CA: cadence; ST: stride time; StPT: stance phase time; SwPT: swing phase time; CV: coefficient of variation; ROM: range of motion; AJ: ankle joint; KJ: knee joint; HJ: hip joint; HS: heel strike angle; TO: toe-off angle; H–Y stage: Hoehn–Yahr stage; UPDRS I: Unified Parkinson’s Disease Rating Scale part 1; UPDRS II: Unified Parkinson’s Disease Rating Scale part 2; UPDRS III: Unified Parkinson’s Disease Rating Scale part 3; PDQ39: Parkinson’s Disease Questionnaire-39.

## Data Availability

The data supporting the findings of this study are included in the article, further inquiries can be directed to the corresponding authors.

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
