# Peer review of "Gait Analysis with Wearables Is a Potential Progression Marker in Parkinson’s Disease"

_brainsci, 2022, doi:10.3390/brainsci12091213_

Round 1

Reviewer 1 Report

The authors have used JiBuEn gait analysis system to investigate the gait deterioration in Parkinson’s Disease (PD) as a potential progression marker of the disease development in PD patients. One hundred thirty eight PD patients and 56 healthy individuals as a control group were included in the study. The authors  conclude that gait parameters may be helpful for the detection of PD and its progression in individuals suffering from PD.

The study is very interesting and it can be beneficial for the healthcare related fields. I do have some comments shown below:

 The name of the gait analysis system is better to be mentioned in line 18 .

Line 222: the heading should be modified.

Conclusion is missing. Parts of Line 223 to 251 may be included in the conclusion section.

Line 241 to 243: sentences should be separated.

Author Response

Point 1: The name of the gait analysis system is better to be mentioned in line 18

Response: Thanks for your hard work. We have added the name of gait analysis system in our manuscript (Line 20)

Point 2: Line 222: the heading should be modified.

Response: Thank the reviewer very much for the comment. After careful inspection, we adjusted the heading to be consistent (Line 215, Line 267).

Point 3: Conclusion is missing. Parts of Line 223 to 251 may be included in the conclusion section.

Response: Thank you for your comment. We have listed the conclusions separately in our manuscript (Line 297-305)

Point 4: Line 241 to 243: sentences should be separated.

Response: Thank you for this comment. We have separated this sentences (292-293).

Reviewer 2 Report

The current research presents a biomechanical evaluation of gait parameters in patients with Parkinson’s disease, related to the different stage of Hoehn and Yahr scale. 138 PD patients and 56 healthy subjects were include in the study. Gait spatio temporal parameters and joints kinematics were considered as output of interest. Results demonstrated that gait impairment occurs in the early stage of PD and deteriorates among the progression of the disease.

Despite the current interest in the proposed topics (PD patients, objective biomechanical measurements, wearable devices) and the promising results obtained from the experimental tests, some important and crucial revisions need to be applied in order to improved the manuscript. The introduction well presents the Parkinson’s disease and the classification based on the clinical scale, but no information about previous biomechanical studies have been reported. Moreover, no introduction about the use of wearable systems in clinical application have been depicted. This section must be deeply improved with the description of previous studies, stressing the limitations and the novelty proposed by the present study. In the methodology section it is necessary to improve the description of the used instrumentation, and a picture of the sensors and of the environment might be helpful for the understanding of experimental set-up. It is not clear how the parameters are measured. Results are well showed with tables and figures and the discussion highlighted a comparison with previous studies. Nevertheless, it is unclear the novelty of the proposed analysis, which depicted the evolution of parameters based on the different stages of the clinical scale. I suggest to revise the statistical analysis with the integration of additional tests with the attempt to underline any correlations between the clinical and instrumental evaluation. It could improve the significance of the study. Moreover, it is important to stress the novelty with a comparison with lack of previous analysis. This part must be improved in relation with the introduction. Finally, the authors declare as limitation the number of the involved patients. In my personal opinion, I don’t think that, due to the large amount of the subjects (138 PD, 56 healthy). Possible limitations might be identified in the adopted instrumentation, setting for the experiments and biomechanical parameters.

For all these reasons, I recommend the reconsideration after major revisions.

Author Response

Point 1: The introduction well presents the Parkinson’s disease and the classification based on the clinical scale, but no information about previous biomechanical studies have been reported. Moreover, no introduction about the use of wearable systems in clinical application have been depicted. This section must be deeply improved with the description of previous studies, stressing the limitations and the novelty proposed by the present study.

Response: Thank you for your hard work. We have We compared some of the sensing technique in the past. We emphasized the advantages of wearables and the novelty of the article in the introduction section (Line57-66).

Point 2: In the methodology section it is necessary to improve the description of the used instrumentation, and a picture of the sensors and of the environment might be helpful for the understanding of experimental set-up. It is not clear how the parameters are measured.

Response: Thank you for this comment. The JiBuEn gait analysis system consists of Bluetooth module fixed under the smart shoes and modules with inertial micro-electro-mechanical system sensors. Moreover, 4 modules tied to the subject’s thigh and calf, and 1 module tied to the subject’s waist. Through wireless Bluetooth transmission technology, the gait data are transmitted into the computer system in real time and the final gait data were obtained. We also added a wearing image of the JiBuEn gait system to the article for easy understanding (Figure 1.).

Point 3: Results are well showed with tables and figures and the discussion highlighted a comparison with previous studies. Nevertheless, it is unclear the novelty of the proposed analysis, which depicted the evolution of parameters based on the different stages of the clinical scale. I suggest to revise the statistical analysis with the integration of additional tests with the attempt to underline any correlations between the clinical and instrumental evaluation. It could improve the significance of the study. Moreover, it is important to stress the novelty with a comparison with lack of previous analysis. This part must be improved in relation with the introduction.

Response: Thank the reviewer very much for the comment. We performed Spearman correlation analysis to investigate the association between gait parameters and H-Y stage, UPDRS scores, and PDQ39. We found that gait parameters are significantly related to H-Y stage, UPDRS â… , UPDRS â…¡, UPDRS â…¢ and PDQ39 (Table 3). We have also added relevant discussions (Line282-289). The correlation between objective gait parameters and these scales associated with disease progression makes gait parameters more significant as objective markers of disease progression in Parkinson's disease

Point 4: Finally, the authors declare as limitation the number of the involved patients. In my personal opinion, I don’t think that, due to the large amount of the subjects (138 PD, 56 healthy). Possible limitations might be identified in the adopted instrumentation, setting for the experiments and biomechanical parameters.

Response: Thank you for your comment. Although our sample size of PD patients is not so small, there are relatively few patients in H-Y stage IV. So we think that's one of our limitations. For the JiBuEn gait analysis system, the zero-correction algorithm, hexahedral calibration technique, and high-order low-pass filter were used in data preprocessing which can decrease high-frequency noise and reduce accumulative errors. The accuracy of this equipment has been verified before. In addition, all participants walked twice in advance to familiarize themselves with the test.
